# Does Secondary Plant Metabolite Ursolic Acid Exhibit Antibacterial Activity against Uropathogenic *Escherichia coli* Living in Single- and Multispecies Biofilms?

**DOI:** 10.3390/pharmaceutics14081691

**Published:** 2022-08-14

**Authors:** Zuzanna Sycz, Dorota Wojnicz, Dorota Tichaczek-Goska

**Affiliations:** Department of Biology and Medical Parasitology, Wroclaw Medical University, 50-345 Wroclaw, Poland

**Keywords:** ursolic acid, plant metabolites, multispecies biofilm, urinary tract infections, uropathogenic bacteria

## Abstract

Multispecies bacterial biofilms are the often cause of chronic recurrent urinary tract infections within the human population. Eradicating such a complex bacterial consortium with standard pharmacotherapy is often unsuccessful. Therefore, plant-derived compounds are currently being researched as an alternative strategy to antibiotic therapy for preventing bacterial biofilm formation and facilitating its eradication. Therefore, our research aimed to determine the effect of secondary plant metabolite ursolic acid (UA) on the growth and survival, the quantity of exopolysaccharides formed, metabolic activity, and morphology of uropathogenic Gram-negative rods living in single- and mixed-species biofilms at various stages of their development. Spectrophotometric methods were used for biofilm mass formation and metabolic activity determination. The survival of bacteria was established using the serial dilution assay. The decrease in survival and inhibition of biofilm creation, both single- and multispecies, as well as changes in the morphology of bacterial cells were noticed. As UA exhibited better activity against young biofilms, the use of UA-containing formulations, especially during the initial steps of urinary tract infection, seems to be reasonable. However, the future direction should be a thorough understanding of the mechanisms of UA activity as a bioactive substance.

## 1. Introduction

Ursolic acid (UA; 3β-hydroxy-12-ursen-28-ic acid) belongs to the group of pentacyclic triterpenes (TPs), polycyclic compounds classified as secondary plant metabolites. Large amounts of UA are found in the leaves of *Arctostaphylos uva-ursi*, known as bearberry. This plant is a pharmacopeial material with antibacterial, anti-inflammatory, and diuretic properties, used in modern phytotherapy as an adjuvant in the treatment of urinary tract infections (UTIs) [1,2,3,4].

The in vitro and in vivo studies to date have confirmed the numerous pharmacological properties of UA, including antineoplastic, hypotensive, cardioprotective, antidiabetic, nephroprotective, diuretic, immunostimulating, antimalarial, anticryptococcal, antituberculous, and antiviral [5,6,7,8,9,10,11,12]. In addition, UA may accelerate the healing process of wounds, stimulate muscle growth, and reduce the growth of adipose tissue, and it is also a component of dermatological preparations, cosmetics, and dietary supplements for athletes [13,14,15].

The experiments carried out in recent years have also shown that UA alone and in combination with antibiotics (including β-lactams, tetracyclines, fluoroquinolones, and aminoglycosides) shows significant antibacterial activity, but these studies mainly concerned bacteria living in suspension—the so-called planktonic forms [16,17,18,19,20,21,22,23,24,25,26,27,28,29,30,31,32,33,34,35,36,37,38,39,40,41,42].

The above studies included the determination of the minimal inhibitory concentration (MIC) value of UA and its influence on the survival of bacteria, on the morphology of cells and bacterial membranes as well as on the impact on bacterial virulence factors such as hydrophobicity of the surface of bacterial cells, ability to move, and synthesis of the curli fimbria and P-type fimbria. Few research centers have investigated the effect of UA on the ability of bacteria to form biofilms, and these studies have focused on single-species consortia [30,41,43,44,45,46,47,48,49,50,51,52,53,54,55,56,57].

It is known that both the biofilms formed by commensal and pathogenic bacteria can constitute single-species consortia but also can consist of many species of microorganisms. Examples of both single- and multispecies structures are formed in the course of a urinary tract infection (UTI). The main etiological factor of UTI is uropathogenic *Escherichia coli*, accompanied by other species, including *Enterobacter cloacae* and *Pseudomonas aeruginosa* [58]. The eradication of such a multispecies biofilm with standard pharmacotherapy is often unsuccessful. Therefore, plant-derived compounds are currently being researched as an alternative strategy to antibiotic therapy for preventing the formation of bacterial biofilm and facilitating its removal. The effect of UA on multispecies biofilms is limited to the influence of this acid on Gram-positive bacteria from the genus *Streptococcus* [59].

Therefore, our research aimed to determine the effect of UA on the growth and survival of *E. coli* rods living in single-, dual- (*E. coli + E. cloacae*; *E. coli + P. aeruginosa*), and triple-species (*E. coli + E. cloacae + P. aeruginosa*) biofilms at various stages of their development. We also attempted to assess the differences in the amount of biofilm mass formed, metabolic activity, and the morphology of bacteria with the treatment of UA in the various biofilm consortia depending on their species content.

## 2. Materials and Methods

### 2.1. Bacterial Strains

Three uropathogenic reference strains from the American Type Culture Collection (ATCC) were used in the study: *Escherichia coli* CFT073 (ATCC 700928), *Enterobacter cloacae* ATCC-BAA 2468, *Pseudomonas aeruginosa* ATCC 25000. The strains were maintained at −80 °C in nutrient broth containing 40% glycerol.

### 2.2. Cultural Media and Chemicals

The following culture media and chemicals were used: chromogenic coliform agar CFS 1424 (Graso Biotech, Starogard Gdański, Poland), tripticasein soy broth (TSB; Biocorp, Warsaw, Poland), tripticasein soy agar (TSA; Biocorp, Warsaw, Poland), nutrient broth (SIS-Biomed, Warsaw, Poland), lab agar (Biocorp, Warsaw, Poland), Mueller–Hinton broth II (MHB; International Diagnostic Group plc, Cardiff, UK), dimethyl sulfoxide (DMSO; Sigma-Aldrich, Poznań, Poland), phosphate-buffered saline (PBS; Polgen, Warsaw, Poland), crystal violet (CV; Sigma-Aldrich, Poznań, Poland), 2,3,5-triphenyl tetrazolium chloride (TTC; Sigma-Aldrich, Poznań, Poland).

### 2.3. Antimicrobial Agent

Ursolic acid (UA, purity ≥ 90%) was purchased from Sigma-Aldrich (Poznań, Poland). Stock solution at a concentration of 4 mg/mL was prepared each time by dissolving UA in DMSO. For further experiments, the final concentration of UA was prepared by diluting the stock with an appropriate culture medium.

### 2.4. Preparation of Bacterial Suspension

The bacteria were suspended in 1 mL of TSB and incubated in a shaking water bath (37 °C for 2 h). After centrifugation (4500 rpm/5 min), a sufficient volume of PBS was added to the pellet to obtain a bacterial suspension with an optical density of 0.5 McFarland (1–2 × 10^8^ CFU/mL).

### 2.5. Preparation of Biofilm Cultures

Bacterial cultures were carried out in 96-well polystyrene microtiter plates for 6, 24, 48, 72, and 96 h. Single-species (*E. coli*; *E. cloacae*; *P. aeruginosa*), dual-species (*E. coli + E. cloacae*; *E. coli + P. aeruginosa*) and triple-species (*E. coli + E. cloacae + P. aeruginosa*) biofilms were prepared. The control biofilms did not contain UA, while the test samples contained UA at a concentration of 0.5 × MIC (768 μg/mL). Twenty microliters of each bacterial suspension with a density of 1–2 × 10^8^ CFU/mL was added to each UA-treated and untreated (control) sample.

### 2.6. Biofilm Formation on Microtiter Plates and Count of Live Bacteria in Biofilm

Bacterial survival in biofilm was established after each time of incubation (6, 24, 48, 72, and 96 h). Biofilm cultures were gently washed away to remove cells not bound to the matrix. The biofilm deposited on the wall of the microplate well was scraped with a sterile spatula [60] and resuspended in PBS. Then, the number of live bacteria in the biofilm (CFU/mL) was determined by serial dilution and plating on agar plates. Three biological replicates were performed, and each had 6 technical replicates. The final results are average values.

### 2.7. Biofilm Formation Assay and Quantification

From 6, 24, 48, 72, and 96 h biofilm cultures, cells unbound with matrix were removed by gentle rinsing three times with sterile distilled water. Then, 1% CV was added to each well and incubated at 37 °C for 15 min to cause the CV to penetrate the biofilm structure. The dye was then removed, and 95% ethanol was added to wash the CV off the biofilm matrix. After 15 min of incubation at room temperature, the optical density (OD) was measured at a wavelength of 590 nm on a microplate reader (HiPo MPP-96^®^ BIOSAN, Riga, Latvia). Based on the OD value, bacterial strains were classified into one of the following groups: OD ≤ ODc—not producing biofilm; ODc < OD ≤ 2 × ODc—weak biofilm-producing; 2 × ODc < OD ≤ 4 × ODc—producing moderate biofilm; 4 × ODc < OD—producing strong biofilm. The ODc cut-off value was calculated as the sum of the mean OD value for the blank (TSB) and 3 times the standard deviation of the mean OD for TSB [61,62,63]. The experiment was repeated three times. The optical density was read from 6 wells of the microtiter plate, giving a total of 18 repeats. The final results are average values.

### 2.8. Determination of the Metabolic Activity of Bacteria in a Biofilm by Spectrophotometry

The 10% TTC was added to 6, 24, 48, 72, and 96 h biofilm cultures. The metabolically active bacteria reduced TTC to red triphenyl formazan. After 24 h of incubation at 37 °C, planktonic cells were removed by rinsing the biofilm cultures three times with sterile distilled water. Then, 95% ethanol was added to each culture, and after 15 min of incubation at room temperature, the amount of produced triphenyl formazan was measured spectrophotometrically (HiPo MPP-96^®^ BIOSAN, Riga, Latvia) at a wavelength of 490 nm [64,65]. The experiment was repeated three times. The optical density was read from 6 wells of the microtiter plate, giving a total of 18 repeats. The final results are average values.

### 2.9. Effect of UA on Bacterial Morphology

Bacterial biofilms were incubated at 37 °C for 6, 24, 48, 72, and 96 h without UA and with UA at a concentration of 0.5 × MIC in 96-well microtiter plates. The samples were then washed three times in PBS. Then, to visualize bacterial cells, 20 μL of each culture was transferred onto a glass slide, air-dried, Gram-stained, and observed in a microscope (Nikon Eclipse 400, Tokyo, Japan) with 1000-fold magnification. The changes in bacterial cell morphology were recorded with the use of PROGRES GRYPHAX^®^ Version 2.2.0.1234, Jenoptik, Jena, Germany.

### 2.10. Statistical Analysis

The nonparametric Kruskal–Wallis test followed by a Dunn’s multiple comparison test was used in analysis of the obtained results. Statistical calculations were performed using Statistica 13.3. (Stat Soft, Kraków, Poland). All values are expressed as mean ± SD. Values of *p* ≤ 0.05 were considered statistically significant.

## 3. Results

### 3.1. Survival of E. coli in Single-, Dual-, and Triple-Species Biofilms

The aim of the current study was bidirectional. First, we decided to evaluate how the presence of the other bacterial species influences the survival, biofilm mass formation, and metabolic activity of the *E. coli* rods—the most prevalent cause of UTIs. The second very important and completely innovative goal of our research was to assess how the bacteria living in single- and multispecies biofilms are affected by the secondary plant metabolite ursolic acid (UA).

Comparing the number of live *E. coli* cells in single- and dual-species biofilms (*E. coli + E. cloacae; E. coli + P. aeruginosa*) untreated with UA, it was found that the presence of both *E. cloacae* and *P. aeruginosa* rods limited the abundance of the *E. coli* strain (Figure 1). A statistically significant reduction in the number of *E. coli* cells (*p* ≤ 0.05) was noted in the *E. coli + E. cloacae* biofilm after 6 and 24 h and in the *E. coli + P. aeruginosa* biofilm at all stages of its formation, except for the young 6 h culture. The data present in Figure 1 also show that the simultaneous presence of *E. cloacae* and *P. aeruginosa* in the triple-species consortium (*E. coli + E. cloacae + P. aeruginosa*) significantly limited the growth of *E. coli* at all stages of its formation compared to the growth of *E. coli* in monoculture (*p* ≤ 0.05).

Data present in Figure 1 show that UA significantly decreased the number of *E. coli* in single-species biofilm regardless of its maturity stage (*p* ≤ 0.05). The greatest reduction in the number of viable cells (2 log_10_) occurred after 6 and 24 h of incubation. In the dual-species consortia (*E. coli + E. cloacae; E. coli + P. aeruginosa*), UA significantly reduced the number of *E. coli* rods after 6, 24, and 48 h of incubation. A significant reduction in *E. coli* survival under the influence of UA was also observed in young 6 and 24 h triple-species biofilms (*E. coli + E. cloacae + P. aeruginosa*) (*p* ≤ 0.05). Unfortunately, in mature 72 and 96 h cultures exposed to UA, of both dual-species (*E. coli + P. aeruginosa*) and triple-species (*E. coli + E. cloacae + P. aeruginosa*), the number of *E. coli* significantly increased in comparison to the samples untreated with UA.

Comparative analysis of the survival of *E. coli* in the tested biofilms exposed to UA (Figure 1) showed that this acid most significantly affected *E. coli* bacteria living in monocultures regardless of the incubation time.

### 3.2. Formation of Biofilm Mass by Uropathogenic Rods in Nontreated (Control) and UA-Treated Samples

Based on the amount of biofilm formed, read as the optical density (OD) value, the tested bacterial strains were classified into one of three groups: non-biofilm-forming OD_590_ ≤ 0.218, weak biofilm-producing 0.218 < OD_590_ ≤ 0.436, and moderate biofilm-forming 0.436 < OD_590_ ≤ 0.872. None of the strains was a strong biofilm producer (OD_590_ > 0.872).

The data in Figure 2 show that the *E. coli* strain formed a weak biofilm at all stages of its development. The largest amount of biofilm was synthesized in a 6 h culture (OD_590_ = 0.406) and the lowest in a 48 h culture (OD_590_ = 0.261).

The *E. cloacae* strain produced a similar amount of the biofilm mass as *E. coli* (0.281 < OD_590_ ≤ 0.329), except for the 24 h culture where it was a better biofilm producer than the *E. coli* strain (OD_590_ = 0.574). The *P. aeruginosa* strain showed the best biofilm production among the three bacterial strains tested. It produced a higher biofilm amount (0.802 < OD_590_ ≤ 0.836) than the two other tested strains at all stages of biofilm formation except for the young 6 h culture (OD_590_ = 0.370).

The analysis of the results presented in Figure 2 shows that the amounts of biofilm mass being produced by the bacteria growing in 24, 48, and 72 h dual-species consortia (*E. coli + E. cloacae*) were smaller than the amounts of the biofilm produced by each of these species when growing separately. However, statistical significance was only demonstrated in the 24 h culture (*p* ≤ 0.05).

A similar result was noted for the *E. coli + P. aeruginosa* dual-species biofilm where it is seen that the presence *of E. coli* antagonistically influences the biofilm mass production by *P. aeruginosa* in the 24, 48, and 72 h cultures (*p* ≤ 0.05). The reductions in biofilm mass in these cultures compared to *P. aeruginosa* monoculture were 34%, 51%, and 30%, respectively. The data contained in Figure 2 show that in 24, 48, and 72 h triple-species biofilms (*E. coli + E. cloacae + P. aeruginosa*), the presence of *E. coli* and *E. cloacae* also significantly reduced the biofilm mass formation by *P. aeruginosa* rods (*p* ≤ 0.05). The reductions in biofilm mass in these cultures were 38%, 52%, and 26%, respectively.

The data presented in Figure 2 show that UA significantly inhibited biofilm production in all mono- and multispecies 6 h cultures (OD_590_ ≤ 0.218) compared to the UA-untreated controls (*p* ≤ 0.05). It was also found that UA had an antibiotic effect on the older 24, 48, 72, and 96 h consortia in which *P. aeruginosa* rods were present (*p* ≤ 0.05). Additionally, in the 24 h *E. cloacae* culture, UA also significantly reduced the amount of the biofilm synthesized by this strain (*p* ≤ 0.05).

### 3.3. Determination of the Metabolic Activity of Bacteria Living in Mono-, Dual- and Triple-Species Biofilms

The results presented in Figure 3 indicate that *E. cloacae* in the 24 h culture and *E. coli* in the 96 h consortium showed the highest metabolic activity. Analyzing the metabolic activity of bacteria growing in the dual-species *E. coli + E. cloacae* biofilm, it was found that their metabolic activity was significantly lower than that of *E. cloacae* growing in monoculture (*p* ≤ 0.05).

The metabolic activity of bacteria growing in 24, 48, and 96 h triple-species biofilms was weaker than the metabolic activity of each of these species of rods growing separately. However, a statistically significant result was only observed for the 24 h *E. cloacae* monoculture (*p* ≤ 0.05).

The data in Figure 4 indicate that UA reduced the metabolic activity of the rods growing in all biofilms tested at all stages of their development except for the young 6-h-old cultures. The UA was the most effective against 24 h consortia. The metabolic activity of rods treated with UA was significantly reduced in both single- and dual-species consortia with a reduction ratio of 28.3–40.5% and 21.1–22.9%, respectively (*p* ≤ 0.05).

In older biofilms, UA was less effective, significantly reducing only the metabolic activity of *E. cloacae* rods after 48 h (25.4% reduction), *E. coli* rods after 72 (18.2%) and 96 h (34.1%), and *E. coli + P. aeruginosa* after 96 h (18.3%) of incubation (*p* ≤ 0.05).

### 3.4. Effect of UA on Bacterial Morphology

Various morphological changes in bacterial cells were observed in the biofilms treated with UA (Figure 5).

In young 6 h single-species cultures formed by *E. coli* and *E. cloacae,* short filaments (5–15 µm) were found. In older biofilms (24–96 h) created by these two species, additionally, long filaments (>15 µm) were noticed (Figure 5A). “Ghost” cells were present in 24, 48, 72, and 96 h *E. coli* monocultures and 48, 72, and 96 h *E. cloacae* single-species biofilms (Figure 5B). In *P. aeruginosa* single-biofilms, short and long filaments were present only in the old 72 and 96 h cultures (Figure 5C). *P. aeruginosa* rods did not form “ghost” cells at any stage of biofilm formation. In young 6 h dual-biofilms formed by *E. coli + E. cloacae* and *E. coli + P. aeruginosa* in the presence of UA, only short filaments were found, and in all older biofilms (24–96 h), long filaments and cells of the “ghost” type (Figure 5D) were additionally observed. In triple-species biofilms (*E. coli + E. cloacae + P. aeruginosa*), short and long filaments were observed regardless of the stage of their development, along with “ghost” cells in 24–96 h consortia (Figure 5E).

## 4. Discussion

### 4.1. Survival of the Escherichia coli Rods in Biofilms Cultured in the Absence of UA

The available literature contains only a few results of studies on multispecies biofilms and the interactions of the bacteria living in them [59,66,67,68,69,70]. The current state of knowledge on the survival of bacteria in biofilms is extended by the current work. It showed that in single-species biofilms, the number of *E. coli* cells is usually greater than in biofilms formed together with other species of bacteria. This indicates the existence of mutually antagonistic interactions between microorganisms. It was noted that the growth of *E. coli* in dual-species consortia was limited by the presence of both *E. cloacae* and *P. aeruginosa*, with *P. aeruginosa* showing a stronger antagonism to *E. coli than E. cloacae*. It is worth noting that the number of *E. coli* cells was the lowest when the rods were grown in a triple-species biofilm.

The antagonism between *E. coli* and *P. aeruginosa* noted by us is also observed by other researchers. Machado et al. [66] investigated 6-day single- and dual-species biofilms composed of *E. coli* K12 MG1655 and *P. aeruginosa* ATCC 10145. In single-species biofilms, the number of *E. coli* cells was greater than in the dual-species consortium. Vanysacker et al. [67] also noticed that in single-species cultures created by *E. coli* LMG 2092T, the number of cells of this strain was greater than in the dual-species consortium created together with *P. aeruginosa* PA14. Similar results were obtained by Cerqueira et al. [68] who observed that the presence of the *P. aeruginosa* in biofilms formed together with *E. coli* rods always adversely affected the count of *E. coli*. Kuznetsova et al. [69] assessed the survival of the *E. coli* strain in monoculture and in a mixed consortium, formed together with one of the three strains of *P. aeruginosa* (the reference ATCC 27853, a clinical strain with high biofilm production and a clinical strain with low biofilm production). The number of *E. coli* cells in the biofilm formed together with *P. aeruginosa* ATCC 27853 did not change from the number of *E. coli* in the monoculture. In contrast, the number of *E. coli* rods decreased when they grew together with the clinical strain of *P. aeruginosa*, regardless of whether it was a weak or a strong biofilm producer.

In contrast to the results obtained by our research team, Oliveira et al. [70] and Solis-Velasquez et al. [71] observed synergism between *E. coli* and *P. aeruginosa*. The number of *E. coli* cells was lower when the strain grew in single-species biofilms than in double-species consortia.

The reasons for the existence of antagonism between *E. coli* and *P. aeruginosa* in dual-species biofilms have not been fully elucidated so far. One of the sources of this phenomenon is quorum sensing (QS), which is one of the main mechanisms involved in biofilm formation. Quorum sensing (QS) is a communication mechanism between bacteria that allows specific processes to be controlled, such as virulence factor expression, biofilm formation, production of secondary metabolites, and stress adaptation mechanisms such as bacterial competition systems, including secretion systems. *P. aeruginosa* has at least three types of QS systems: Las, Rhl, and *Pseudomonas* quinolone signal (PQS). N-(3-oxo-dodecanoyl)-L-homoserine lactone (3-oxo-C12-HSL), N-butanoyl-L-homoserine lactone (C4-HSL), and 2-heptyl-3-hydroxy-4-quinolone (PQS) are used as autoinducers (AIs) in these systems, respectively. Two N-acylhomoserine lactones (AHLs), 3-oxo-C12-HSL and C4-HSL, bind LuxR-type intracellular receptor proteins LasR and RhlR, respectively. In the Las system, a receptor LasR is activated by its cognate 3-oxo-C12-HSL, synthesized by LasI, and this complex activates Las, Rhl, and PQS systems. The Rhl system consists of RhlR and its cognate signal, C4-HSL, synthesized by RhlI, and this complex autoregulates the Rhl system. The PQS system binds its receptor, PqsR, and the complex regulates PQS and Rhl systems. These three QS systems control the production of virulence factors: exoenzymes, exotoxin A, phenazine, lectin, pyocyanin, rhamnolipids (RHLs), fatty acids (cis-2-decenoic acid and cis-11-methyl-2-dodecenoic acid), iron chelating activity, antibiotic resistance, and extracellular polysaccharides (alginate, Psl, and Pel) involved in biofilm formation [72,73]. Due to QS, bacteria not only regulate the number of cells within their own species but also influence the number of other species cells living in a common consortium. Mirani et al. [74] observed that in a young 24 h dual-species consortium, *E. coli* outnumbered *P. aeruginosa*. However, the situation was reversed in the older 48 h biofilm. Only cis-2-decenoic acid (CDA) producing strains of *P. aeruginosa* have been shown to have an antagonistic effect on *E. coli*. CDA belongs to the group of diffusive molecules involved in interspecies signaling and modulating the behavior of other microorganisms. Interestingly, CDA molecules also had the ability to scatter biofilm formed by *E. coli.* Additionally, it was noted that *P. aeruginosa* produced more CDA when grown in with *E. coli* in a mixed biofilm. Rahmani-Badi et al. [75] reported that pure CDA has also been able to inhibit the production of mono- and dual-species biofilms formed by *E. coli* ATCC 25922 and *K. pneumoniae* ATCC 700603 and, moreover, caused the dispersion of biofilms already formed by these bacteria.

In addition to CDA production, *P. aeruginosa* secretes significant amounts of extracellular rhamnolipids (RHLs) and signaling molecules involved in QS, e.g., N-acyl-L-homoserine lactone (AHL) and 2-alkyl-4- quinolones (AQs). *P. aeruginosa* rods, which live in mixed biofilms, use RHL to disperse the biofilm matrix, making it easier for cells to access iron and oxygen for which they compete with other microorganisms. Additionally, AHL molecules activate *P. aeruginosa* to intense divisions, which makes this species dominant in mixed biofilms [76,77]. Cao et al. [78], examining 8–96 h biofilms of *E. coli* DH5α and *P. aeruginosa* PAO1C, observed that *P. aeruginosa* PAO1C strongly secreted AQ molecules in response to the presence of *E. coli* DH5α cells. Interestingly, the appearance of AQ on the periphery of an *E. coli* colonies occurred much earlier than the physical fusion of the colonies of these two species of bacteria.

The importance of extracellular metabolites in the interaction of *E. coli* and *P. aeruginosa* has been confirmed by the studies of Lopes et al. [79]. The authors determined the effect of supernatants containing metabolites of the planktonic and biofilm forms of *P. aeruginosa* ATCC 10145 on the growth of both planktonic and biofilm forms of *E. coli* K12 MG1655. It was shown that the supernatant obtained from the planktonic culture of *P. aeruginosa* did not affect either the biofilm production by *E. coli* or the survival of these rods. On the other hand, metabolites present in the supernatant obtained from *P. aeruginosa* biofilm culture strongly inhibited the development of biofilm formed by *E. coli*. They also significantly reduced the survival of *E. coli* rods.

As demonstrated by Bhattacharjee et al. [80], the more intense growth of *P. aeruginosa* in mixed biofilms, formed together with *E. coli*, is related not only to QS but also to the topography of the surface on which the biofilm grows. The predominance of *P. aeruginosa* only took place when this dual-species cultivation was carried out on a flat surface. During the growth of biofilm on folded surfaces, *E. coli* bacilli were less dispersed. This phenomenon is related to the production of indole by *E. coli* cells, which as a consequence suppresses the dispersion response of these rods to signaling compounds secreted by *P. aeruginosa*. Moreover, Chu et al. [81] found that indole plays a significant role in *E. coli* survival in mixed biofilm by inhibiting the production of pyocyanin (PCN) and other AHL-regulated *P. aeruginosa* virulence factors.

As explained through QS, *P. aeruginosa* secretes antimicrobial chemicals and signal molecules to compete and/or cooperate with other microbes. It is known that the type VI secretion system (T6SS), present in about 25% of Gram-negative bacteria, is often crucial for their virulence and is involved in bacterial interaction and competition, biofilm formation, and transport of ions [72,82,83]. Basler et al. [84] discovered, that the T6SS antibacterial activity of *P. aeruginosa* is triggered by the activity of T6SS displayed by other bacteria (*Vibrio cholerae* and *Acinetobacter baylyi*). *E. coli* and *E. cloacae* also have the T6SS system, but so far it has not been as thoroughly understood as in *P. aeruginosa* [85,86,87]. Soria-Bustos et al. [85] reported that opportunistic strain *E. cloacae* ATCC 13047 codes two functional T6SS systems which are involved in the pathogenesis of *E. cloacae* with specialized functions in the interaction with other bacteria and with host cells.

It can be presumed with high probability that above-described T6SS plays an important role in the antagonism between bacteria in the dual- and especially triple-species consortia we have investigated in this study.

### 4.2. Biofilm Formation and Metabolic Activity of Bacteria Growing in the Absence of UA

The current research showed that both the *E. coli* strain and *E. cloacae* strain formed weak single-species biofilms, while the *P. aeruginosa* strain was distinguished by moderate biofilm production. Wang et al. [88] also noticed that the clinical strains of *P. aeruginosa* they studied produced a much greater amount of biofilm mass than *E. coli* or *E. cloacae*. We also showed that in single-species consortia, the amount of the biofilm mass was usually greater than in dual- and triple-species consortia. The fact that the amount of biomass produced in mature mixed biofilms was lower than the amount of biomass produced at the same stage of biofilm development by each species separately proves the existence of antagonistic interactions between the studied strains of *E. coli*, *E. cloacae*, and *P. aeruginosa*. This phenomenon was particularly evident in dual- and triple-species biofilms, where the amount of the biofilm produced by *P. aeruginosa* was significantly reduced in the presence of *E. coli.*

These results, however, contradict the observations of other authors [69,89,90]. Qian et al. [89] observed that the amount of biofilm produced in single-species consortia of *E. coli* ATCC 25922 and *E. cloacae* ATCC 13047 was lower than in the mixed biofilm formed by both of these species. In the studies by Kuznetsova et al. [69], it was also noticed that the amounts of biomass of the dual-species biofilm composed of *E. coli* and *P. aeruginosa* after 6, 12, and 24 h of incubation were significantly higher in comparison to single-species biofilms. Likewise, Culotti et al. [90] noted that the growth and synthesis of biofilm mass by the *E. coli* DH5α strain was more intense when these bacteria were grown together with *P. aeruginosa* PAO1. In turn, Machado et al. [66] found that the amount of biofilm formed by *E. coli* K12 MG1655 and *P. aeruginosa* ATCC 10145 was similar, regardless of whether the bacteria grew in monocultures or formed a common consortium.

Machado et al. [66] also determined the metabolic activity of *E. coli* and *P. aeruginosa* strains in single- and dual-species biofilms. *P. aeruginosa* in monoculture was characterized by a higher metabolic activity than *E. coli* rods. This result is inconsistent with the results of our current research. *P. aeruginosa* tested by us showed weaker metabolic activity compared to *E. coli* as well as *E. cloacae*. However, both our research and that conducted by Machado et al. [66] showed that bacteria growing in monocultures have a higher metabolic activity than those growing in mixed consortia. Interestingly, our own research showed that the amount of the created biofilm mass did not correlate with the metabolic activity of bacteria. *E. cloacae* and *E. coli* strains producing only weak biofilm were characterized by higher metabolic activity than *P. aeruginosa*, which was a better biofilm producer. The metabolic activity of the bacteria living in biofilm consortia is difficult to interpret, and this is a limitation of the current study. It is known that not all the cells in a biofilm have the same metabolism. The cells living closer to the surface have different metabolism, nutrient, and oxygen availability than the cells inside the biofilm [91].

### 4.3. Survival, Biomass Formation, and Metabolic Activity of Bacteria in Biofilms Treated with UA

The results obtained in our own research showed the antibacterial effect of UA on the rods growing both in monocultures and in mixed biofilms. However, the reduction in the number of viable cells depended on the duration of action of UA and whether the acid was acting on a single- or multispecies consortium. UA reduced the number of *E. coli* rods the most in young 6 and 24 h biofilms, regardless of whether the rods were grown alone or in mixed biofilms.

It is worth noting that UA was most effective in reducing the formation of biofilm mass by *P. aeruginosa* growing in single-species biofilms, as well as by bacteria growing in mixed dual- and triple-species cultures in which *P. aeruginosa* was present. UA also reduced the metabolic activity of bacteria living in single-species biofilms at all stages of their growth, except for the 6 h cultures, in which an increase in bacterial activity was noticed.

Our own previous research also demonstrated the antibacterial activity of UA in the formation and eradication of single-species biofilms formed by the uropathogenic reference *E. coli* CFT073 and clinical *E. coli* strains. UA has also been shown to support the pharmacological effect of ciprofloxacin in removing mature *E. coli* biofilm from urological catheters [53].

The antibacterial activity of UA limited to single-species biofilms of *E.coli* and *P. aeruginosa* was also investigated by others [30,43,50,51]. Ren et al. [43] reported that UA showed significant antibacterial activity against 24 h single-species consortia formed by *E. coli* and *P. aeruginosa* strains, reducing the amount of biofilm mass by 72% and 87%, respectively. Interestingly, UA did not inhibit the growth of these strains growing in planktonic forms but induced the expression of genes encoding proteins related to chemotaxis, e.g., motAB. It is worth noting that overexpression of the motAB gene makes cells too mobile to remain stable in the biofilm environment, which disrupts its formation. Research by Lou et al. [51] also confirmed the antibiotic activity of UA against the strain *P. aeruginosa* ATCC 9027. In turn, Kurek et al. [30] showed that UA had a weak effect on single-species biofilms formed by *P. aeruginosa* strains. However, when combined with β-lactam antibiotics, its antibiotic effect was improved.

Meanwhile, the results obtained by Gilabert et al. [50] did not confirm the antibiotic activity of UA against the strain *P. aeruginosa* ATCC 27853. UA did not inhibit the production of biofilm mass and also stimulated *P. aeruginosa* cells to divide, increasing the survival of these bacilli. It is worth noting, however, that despite the lack of anti-biofilm properties, UA reduced the activity of LasB elastase produced by *P. aeruginosa.* LasB affects the architecture and functionality of the biofilm [92], and inhibition of the activity of this enzyme weakens bacterial adhesion, microcolony formation, and extracellular matrix binding in the biofilm [93].

In the available literature, the influence of UA on multispecies biofilms is limited only to Gram-positive streptococci; therefore, the results of our current research are quite difficult to discuss. However, there are studies in which it has been shown that non-UA antibacterial agents exhibit better activity against single- than multispecies biofilms. Schwering et al. [94] found that *E. coli* and *E. cloacae* biofilms are up to 300 times more resistant to chlorine than single-species consortia. Qian et al. [89] studied the effect of luteolin on the mixed biofilm formed by *E. coli* ATCC 25922 and *E. cloacae* ATCC 13047. They showed that the compound was much less active than when the bacteria grew in monocultures. Interestingly, in our study, UA did not inhibit the survival of *E.coli* in the old 72 and 96 h consortia in which *P. aeruginosa* strain was present. The reason for this could be the presence of *P. aeruginosa* in the biofilm, which over time produces more and more extracellular substances as part of the biofilm matrix. This would make it difficult for UA to access *E. coli* cells grown in such a consortium. The sensitivity of these rods to UA decreases, which in turn causes an increase in the number of *E. coli* cells in the population. However, in a dual- and triple-species consortium untreated with UA, a decrease in the number of *E. coli* cells has been noticed. Thus, perhaps the presence of UA in the biofilm consortium interferes with the signaling molecules involved in QS and the T6SS system described above, and this weakens antagonistic interactions between strains, allowing the divisions of *E. coli* cells.

It is also difficult to discuss the results obtained by us describing the effect of UA on the amount of biomass formed and the metabolic activity of bacteria living in biofilm consortia due to the lack of literature data. Nostro et al. [95] noted changes in the amount of biofilm formed and the metabolic activity of *E. coli* and *Staphylococcus aureus* forming a dual-species consortium in the presence of plant-derived compounds—citronellol and eugenol. Citronellol showed a better effect on single-species biofilms by reducing the amount of biofilm mass and weakening the metabolic activity of bacteria. In contrast, eugenol performed better on the bispecific consortium. A significant decrease in the number of viable cells and their metabolic activity was noticed only in 96 h biofilms. In the present study, UA was shown to be more effective on young 6 and 24 h biofilms.

### 4.4. Changes in Cell Morphology of Bacteria Treated with UA

In biofilm cultures of reference *E. coli* CFT073 (ATCC 700928), *E. cloacae* ATCC-BAA 2468, *P. aeruginosa* ATCC 25000 strains treated with UA, short and long filaments and “ghost” cells partially devoid of the cell wall were observed. Similar morphological cell changes in clinical *E. coli* strains were observed [32]. The authors also found the presence of thickened cells with intracellular distension of the “swollen” type [32]. Wojnicz et al. [41] examined the effect of UA on biofilms formed by clinical strains of *Enterococcus faecalis*. The cells of these cocci had a larger diameter and formed irregular aggregates instead of the characteristic chains visible in UA-untreated cultures. The morphological changes in bacterial cells under the influence of UA were also observed by others. Kurek et al. [23] found that after 24 h of incubation in the presence of UA, the length of *Listeria monocytogenes* bacterial cells was reduced by 20% compared to control. Catteau et al. [39] showed that UA, like the β-lactam antibiotic oxacillin, caused the delocalization of the penicillin-binding protein (PBP2) from the site of the division septum and its redistribution throughout the cell membrane, which disrupted *S. aureus* cell division.

It is known that morphological changes in bacterial cells are conditioned by various mechanisms. The causes of cell filamentation can be both disturbances in peptidoglycan synthesis and disturbances in the formation of division septa. The enzyme PBP3 plays an important role in the formation of septa. Inhibition of its activity causes the cell to lengthen without dividing it. Such cell division inhibiting activity is shown by β-lactam antibiotics having an affinity for PBP3 [96,97]. Filamentation is also observed due to inhibition of replication caused by DNA damage or dysfunction of FtsZ, a key protein in the bacterial division. Septa formation is delayed due to the disfunction of the FtsZ protein by the SulA protein. This stops the division ring formation and inactivates PBP3. Mizushina et al. [98], however, found that UA does not bind directly to DNA and does not affect the activity of *E. coli* DNA polymerase I. Cell filamentation has also been observed under the influence of ciprofloxacin, which by inhibiting the activity of gyrase contributes to blocking DNA replication [99,100]. The formation of “ghost” cells in which partial lysis of the cell wall has taken place is caused both by inhibitors of peptidoglycan synthesis and DNA synthesis [101]. Perhaps UA’s mechanism of action is similar. However, it has not been fully understood and described so far. The morphological changes we observe in the cells of the examined rods after exposure to this acid allow us to assume that it not only destroys the integrity of the cell membrane and causes its dysfunction but can also penetrate into the bacterial cell and interact with DNA or proteins involved in the formation of a division septum. In this way, UA could interfere with the replication process and bacterial cell divisions. Phenotypically altered bacteria may therefore lose their ability to adhere to the host cells, which in turn reduces their virulence.

## 5. Conclusions

Our research has shown a significant effect of UA on the survival of bacterial cells, their morphology, the ability to form single- and mixed-species biofilms, and the metabolic activity of cells living in them. The future direction should be a thorough understanding of the antibacterial mechanism of the activity of UA as a documented bioactive substance. From the perspective of further research, the synergistic effect of UA with antibiotics should also be taken into account. Therefore, it is necessary to define the rules for the validation of the antibacterial activity of UA and the conversion of its in vitro potency into its in vivo therapeutic effect.

Due to the fact that UA exhibited the best antibacterial activity in the early stages of biofilm formation, the use of preparations containing UA would be appropriate, especially at the beginning of UTIs.

## Figures and Tables

**Figure 1 pharmaceutics-14-01691-f001:**
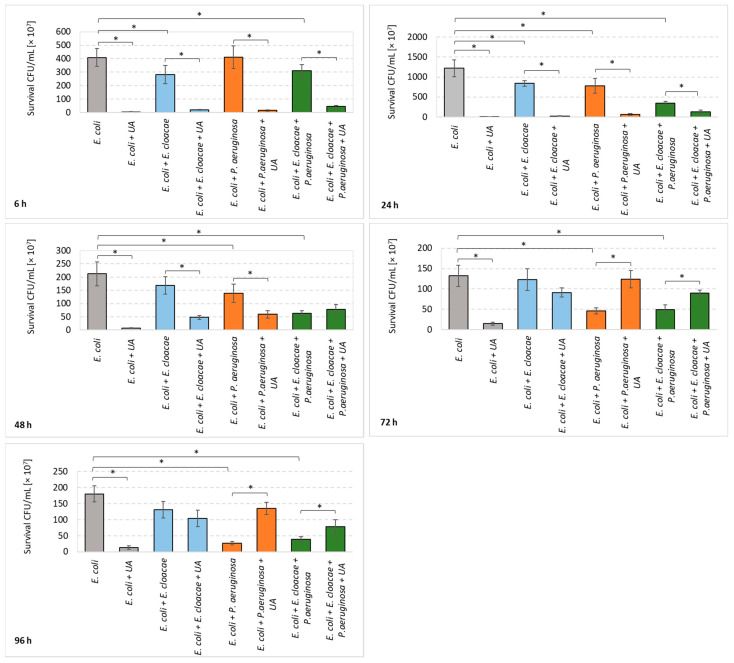
Count of *Escherichia*
*coli* cells living in single-, dual-, and triple-species biofilms untreated (control samples) and treated with ursolic acid (UA). Statistically significant differences (*p* ≤ 0.05) were noted with an asterisk (*). The experiment was repeated three times. The grown bacterial colonies were counted from six plates which gives a total of 18 repeats. The final results are average values.

**Figure 2 pharmaceutics-14-01691-f002:**
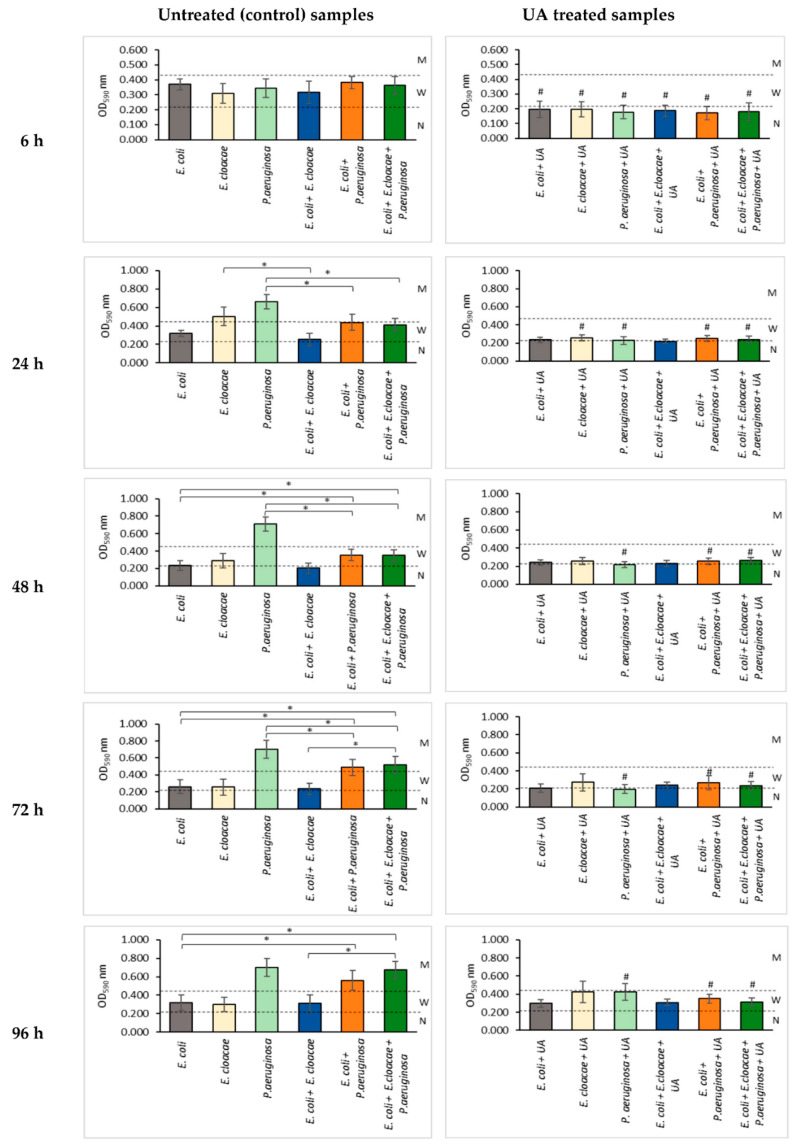
The impact of ursolic acid (UA) on biofilm mass production by bacteria living in single-, dual-, and triple-species biofilms. Statistically significant differences (*p* ≤ 0.05) were noted with an asterisk (*) when untreated samples were compared with each other or with a hash (#) when UA-treated samples were compared to their untreated controls. N—no biofilm; W—weak biofilm; M—moderate biofilm. The experiment was repeated three times. The optical density was read from six wells of the microtiter plate, giving a total of 18 repeats. The final results are average values.

**Figure 3 pharmaceutics-14-01691-f003:**
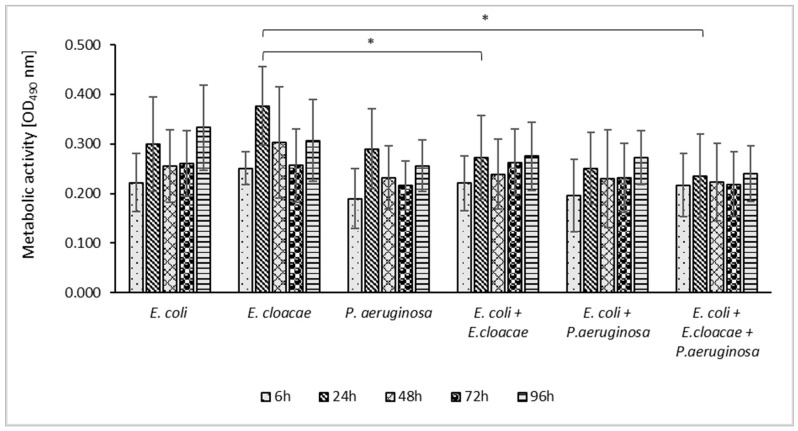
Metabolic activity of bacteria living in mono-, dual- and triple-species biofilms at different stages of biofilm development. Statistically significant differences (*p* ≤ 0.05) were noted with an asterisk (*). The experiment was repeated three times. The optical density was read from six wells of the microtiter plate, giving a total of 18 repeats. The final results are average values.

**Figure 4 pharmaceutics-14-01691-f004:**
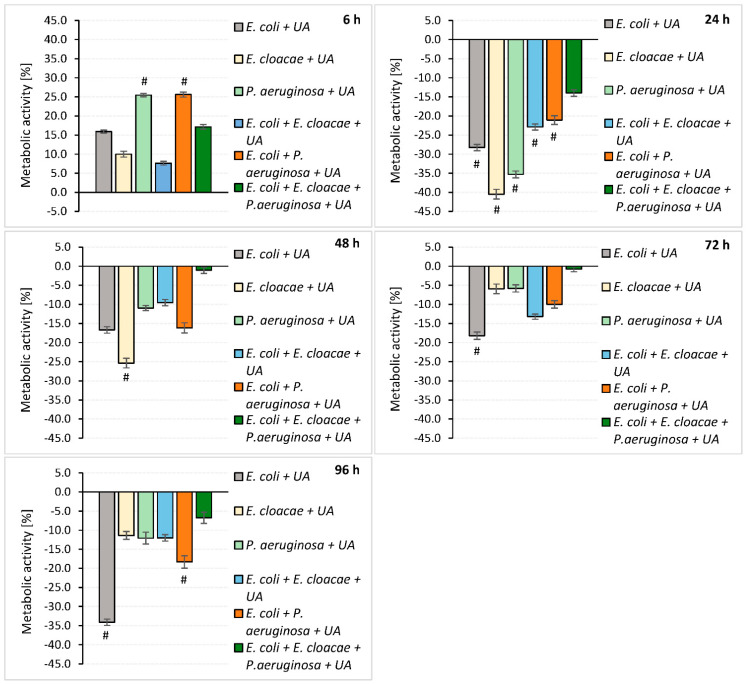
The impact of ursolic acid (UA) on the metabolic activity of bacteria living in mono-, dual- and triple-species biofilms. Statistically significant differences (*p* ≤ 0.05) between UA-treated and untreated samples were noted with a hash (#). The experiment was repeated three times. The optical density was read from six wells of the microtiter plate, giving a total of 18 repeats. The final results are average values.

**Figure 5 pharmaceutics-14-01691-f005:**
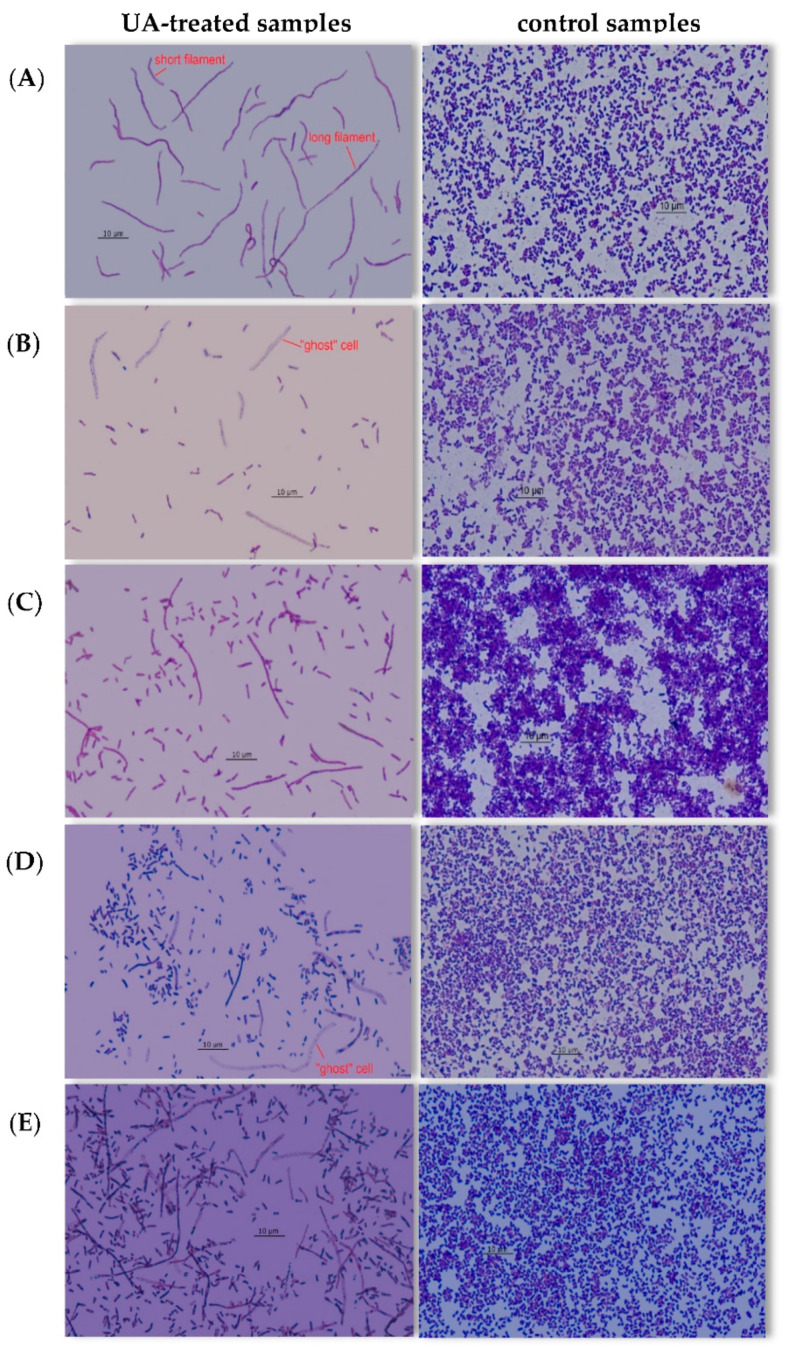
Sample photos showing the morphological changes of bacterial cells exposed to ursolic acid (UA) in different biofilm cultures at different stages of their development; (**A**) 96 h *E. coli* cultures; (**B**) 72 h *E. cloacae* cultures; (**C**) 96 h *P. aeruginosa* cultures; (**D**) 72 h *E. coli + P. aeruginosa* cultures; (**E**) 96 h *E. coli + E. cloacae + P. aeruginosa* cultures; (Nikon Eclipse 400; magnification, ×1000).

## Data Availability

The data presented in this study are available on request from the corresponding author.

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
