# Peer review of "Does Secondary Plant Metabolite Ursolic Acid Exhibit Antibacterial Activity against Uropathogenic Escherichia coli Living in Single- and Multispecies Biofilms?"

_pharmaceutics, 2022, doi:10.3390/pharmaceutics14081691_

Round 1

Reviewer 1 Report

The authors investigate the use of a secondary plant metabolite (ursolic acid) as a novel antibacterial agent. The studies presented are beyond the scope for journal "Pharmaceutics" and are not of general interest for readers of Pharmaceutics. However, the manuscript might be suitable for publication in MDPI Antibiotics or MDPI Bacteria. 

Author Response

The authors investigate the use of a secondary plant metabolite (ursolic acid) as a novel antibacterial agent. The studies presented are beyond the scope for journal "Pharmaceutics" and are not of general interest for readers of Pharmaceutics. However, the manuscript might be suitable for publication in MDPI Antibiotics or MDPI Bacteria.

Authors’ response: thank you for the suggestion

Reviewer 2 Report

The manuscript by Zuzanna Sycz and coworkers deals with the antibacterial properties of ursolic acid, with a focus on its activity against multispecies biofilms. The manuscript is well written and concise. The aim is clear and the concusions sound. The results are supported by the experimental data.

The main concern is about the following statement, that must be discussed in session 4. Discussion:

p.4 lines 161-164: "Unfortunately, in mature 72- and 96-hour cultures exposed to UA, [...] the number of E. coli significantly increased in comparison to the samples untreated with UA." This statement is of importance and should be accompanied by further comments. Such a behaviour is not uncommon and hypothesis on its causes can be drawn, based on literature data.

A few, minor points to be addressed are listed below:

p. 2 lines 61-62 (and p. 14, lines 430-431): "The effect of UA on multispecies biofilms has not been described so far." This statement should take into account the recent paper by Lyu et coworkers, although it is limited to Streptococcus strains:

Lyu X, Wang L, Shui Y, Jiang Q, Chen L, Yang W, He X, Zeng J, Li Y. Ursolic acid inhibits multi-species biofilms developed by Streptococcus mutans, Streptococcus sanguinis, and Streptococcus gordonii. Arch Oral Biol. 2021 May;125:105107. doi: 10.1016/j.archoralbio.2021.105107. Epub 2021 Mar 15. PMID: 33735629.

The above reference can be cited also at p. 11, lines 281-282, where the statement: "The available literature contains only a few results of studies on multispecies biofilms and the interactions of the bacteria living in them." needs literature support.

p.4 lines 166-167: "...showed that this acid affected the most effectively the E. coli bacteria living in 166 monocultures in all of their maturity stages." Statement not clear, please rephrase it.

Author Response

Dear Reviewer,

thank you very much for your valuable comments and suggestions. We took all of them into account. All the introduced changes are in red. Thank you for your time spent reviewing our manuscript.

Comment 1:

Discussion: p.4 lines 161-164: "Unfortunately, in mature 72- and 96-hour cultures exposed to UA, [...] the number of E. coli significantly increased in comparison to the samples untreated with UA." This statement is of importance and should be accompanied by further comments. Such a behaviour is not uncommon and hypothesis on its causes can be drawn, based on literature data.

Authors’ response: thank you, we added the comment to the Discussion section – lines: 600-610

A few, minor points to be addressed are listed below:

Comment 2:

2 lines 61-62 (and p. 14, lines 430-431): "The effect of UA on multispecies biofilms has not been described so far." This statement should take into account the recent paper by Lyu et coworkers, although it is limited to Streptococcus strains: Lyu X, Wang L, Shui Y, Jiang Q, Chen L, Yang W, He X, Zeng J, Li Y. Ursolic acid inhibits multi-species biofilms developed by Streptococcus mutans, Streptococcus sanguinis, and Streptococcus gordonii. Arch Oral Biol. 2021 May;125:105107. doi: 10.1016/j.archoralbio.2021.105107. Epub 2021 Mar 15. PMID: 33735629.

Authors’ response: thank you, the paper by Lyu et al. has been added and the sentence reorganized – line 63

Comment 3:The above reference can be cited also at p. 11, lines 281-282, where the statement: "The available literature contains only a few results of studies on multispecies biofilms and the interactions of the bacteria living in them." needs literature support.

Authors’ response: thank you, the paper by Lyu et al. and the other references have been added – line 361

Comment 4:

p.4 lines 166-167: "...showed that this acid affected the most effectively the E. coli bacteria living in  monocultures in all of their maturity stages." Statement not clear, please rephrase it.

Authors’ response: thank you, the sentence has been corrected – line 181

Reviewer 3 Report

Sycz et al present an interesting study looking at the antibiotic and antibiofilm properties of ursolic acid using 3 medically relevant Gram-negative bacteria (Escherichia coli, Enterobacter cloacae, and Pseudomonas aeruginosa).  There are many strengths of this study, including that they used 3 different bacteria and are looking at viability in both monospecies and multispecies biofilms.  This is an important area of study as many infections are caused by biofilms (formed either by a single bacteria or polymicrobial) and we rapidly need new antimicrobial therapeutics.  Natural products like UA are promising candidates for use as antimicrobials by themselves or in conjuction with existing antibiotics.  However, there were some problems with the methods and conclusions as described below.

Line 38 (minor comment):  references are needed at the end of this sentence (unless all the references also refer to the next sentence)

Line 56 (minor comment):  a slight misrepresentation of UTIs as many are monomicrobial/monospecies infections

Methods:  how were bacterial cultures stored?  Frozen at -80 oC?  How many cells were used to inoculate the biofilm assays?

Methods for bacterial morphology assays:  More details are needed here.  How were the cells prepared for this?  The methods say that they were grown as biofilms but the final pellets were dried on slides.  Were the biofilms grown on glass slides?  Or were the biofilms grown in something else, then removed, then placed on glass slides?  If so, wouldnt the morphology be damaged by transferring to the glass slides?  Did the drying process affect the cell morphology?  Also, what software was used to do the microscopy and process the images?

Methods:  how many replicates were done for each assay?  This needs to be stated for each type of experiment.  It would be helpful to include this information in the figure legends as well as the methods section.

Line 142:  this is overstating the importance of this study.  While other studies may not have looked at antibiofilm activities of this exact compound, many other studies have investigated antibiofilm properties of other natural products.   This is still an exciting study.

Did the authors control for growth when analyzing the biofilm data?  This is commonly done by dividing the OD590 value by the OD600 value as this tells you how much biofilm was made per cell.  Since the co-cultures and triple cultures had lower survival than just E. coli by itself, a similar OD590 value across all these cultures actually suggests the co-cultures and triple cultures are producing more biofilm material per cell.

I appreciate that the authors wanted to use cutoffs for low, medium, and high biofilm but the numbers really only apply to this study.  What is a high biofilm here might be a low biofilm when looking at other clinical isolates.  It would be more meaningful to just qualitatively compare the levels of biofilm between the strains (Pseudomonas produced more biofilm than E. coli, etc).

Line 207 and 219:  the authors did not directly measure the EPS production of any strains, only quantified the overall biofilm production.  Crystal violet will stain cells as well as biofilm matrix (EPS).  Therefore it does not seem correct to say that any strain influenced the EPS production of another strain and the description of the data should instead be limited to talking about overall biofilm levels.  EPS is also talked about in the discussion and this should also be modified since the authors are not measuring EPS.

It is possible that the metabolic biofilm data is difficult to interpret because not all cells in a biofilm have the same metabolism (this has been shown previously in Pseudomonas – the cells closer to the surface have different metabolism, oxygen content, etc than the cells further away closer to the surface of the biofilm).  The authors should comment on this as a limitation of this assay.

Figure 4:  there are no error bars on this figure.  How many replicates were done?

Figure 5:  the images here are very nice, but the controls were not appropriately done.  For each UA treated sample there needs to be an untreated (no UA) sample of the same bacteria at the same time point.  The only untreated (no UA) sample is at 48 hr but none of the other panels show samples taken at 48 hr.

Paragraph starting line 266:  not all of the data is shown in Figure 5 so it is difficult to interpret some of this data (6 hour single species cultures, ghost cells shown in monocultures at different time points, etc).  If the authors have images from all time points (which presumably they do since they discuss what the cells looked like) then it would be helpful to add these to the figure to show the complete morphological patterns.

Discussion:  the type 6 secretion system is a major mechanism for antagonism between Gram-negative bacteria.  Pseudomonas and E. cloacae both encode T6SS genes.  Could the authors please comment on whether they think inhibition mechanisms like T6SS are responsible for the decreased viability in the triple species cultures?

Author Response

Dear Reviewer,

thank you very much for your valuable comments and suggestions. We took all of them into account. All the introduced changes are in red. Thank you for your time spent reviewing our manuscript.

Comment 1:

Line 38 (minor comment):  references are needed at the end of this sentence (unless all the references also refer to the next sentence)

Authors’ response: thank you for the suggestion, references have been added at the end of the sentence – lines 38, 41

Comment 2:

Line 56 (minor comment):  a slight misrepresentation of UTIs as many are monomicrobial/ monospecies infections

Authors’ response: thank you, the sentences (lines 53-56) have been changed

Comment 3:

Methods:  how were bacterial cultures stored?  Frozen at -80 oC?  How many cells were used to inoculate the biofilm assays?

Authors’ response: information about storage and bacterial cells number which were used to inoculate biofilm assay have been added to the text – lines 74-75 and 94, 100-102.

Comment 4:

Methods for bacterial morphology assays:  More details are needed here.  How were the cells prepared for this?  The methods say that they were grown as biofilms but the final pellets were dried on slides.  Were the biofilms grown on glass slides?  Or were the biofilms grown in something else, then removed, then placed on glass slides?  If so, wouldn’t the morphology be damaged by transferring to the glass slides?  Did the drying process affect the cell morphology?  Also, what software was used to do the microscopy and process the images?

Authors’ response: thank you, more details for the description of the bacterial morphology assay have been added to the text – lines 139-144

We also added the name of the software used to do the microscopy and process the images PROGRES GRYPHAX® – line 144

Neither the transfer of bacterial suspensions nor the air-drying process does damage the bacterial cells and affects their morphology.

 Comment 5:

Methods:  how many replicates were done for each assay?  This needs to be stated for each type of experiment.  It would be helpful to include this information in the figure legends as well as the methods section.

Authors response: thank you, the requested information has been added to each assay description as well as the figure legend

Comment 6:

Line 142:  this is overstating the importance of this study.  While other studies may not have looked at antibiofilm activities of this exact compound, many other studies have investigated antibiofilm properties of other natural products.   This is still an exciting study.

Authors response: Thank you for appreciating the value of our study. In our opinion, it is really innovative and valuable, as there is no research on the influence of UA on multispecies biofilms except for one paper by Lyu et al (2021) which has been added to the manuscript.

Comment 7:

Did the authors control for growth when analyzing the biofilm data?  This is commonly done by dividing the OD590 value by the OD600 value as this tells you how much biofilm was made per cell.  Since the co-cultures and triple cultures had lower survival than just E. coli by itself, a similar OD590 value across all these cultures actually suggests the co-cultures and triple cultures are producing more biofilm material per cell.

Authors’ response: Thank you, this is a fair point. Unfortunately, the growth control we carried out consisted only in counting the colonies grown on the chromogenic agar. This was done in parallel with each biofilm assay. Determining the OD600 values would give an insight into the amount of biofilm produced per single bacterial cell.  We will take this valuable suggestion into account in our future research.

Comment 8:

I appreciate that the authors wanted to use cutoffs for low, medium, and high biofilm but the numbers really only apply to this study.  What is a “high” biofilm here might be a “low” biofilm when looking at other clinical isolates.  It would be more meaningful to just qualitatively compare the levels of biofilm between the strains (Pseudomonas produced more biofilm than E. coli, etc).

Authors’ response: The criteria used in this study are used by many authors. Nevertheless, this suggestion is absolutely correct, because what in this study is considered a moderate biofilm, in fact in the next study may be a weak or strong biofilm. Accordingly, we have modified the text of the manuscript. We hope that it is now more readable and valuable.

Comment 9:

Line 207 and 219:  the authors did not directly measure the EPS production of any strains, only quantified the overall biofilm production.  Crystal violet will stain cells as well as biofilm matrix (EPS).  Therefore it does not seem correct to say that any strain influenced the EPS production of another strain and the description of the data should instead be limited to talking about overall biofilm levels.  EPS is also talked about in the discussion and this should also be modified since the authors are not measuring EPS.

Authors’ response: thank you for this valuable suggestion, the text has been corrected – lines 219-238

Comment 10:

It is possible that the metabolic biofilm data is difficult to interpret because not all cells in a biofilm have the same metabolism (this has been shown previously in Pseudomonas – the cells closer to the surface have different metabolism, oxygen content, etc than the cells further away closer to the surface of the biofilm).  The authors should comment on this as a limitation of this assay.

Authors’ response: thank you for this valuable suggestion, the comment has been added to the Discussion section – lines 548-552

Comment 11:

Figure 4:  there are no error bars on this figure.  How many replicates were done?

Authors’ response: thank you, this is our oversight. Error bars have been added; the experiment was done three times.

Comment 12:

Figure 5:  the images here are very nice, but the controls were not appropriately done.  For each UA treated sample there needs to be an untreated (no UA) sample of the same bacteria at the same time point.  The only untreated (no UA) sample is at 48 hr but none of the other panels show samples taken at 48 hr.

Authors ‘response: Thank you, the sample images of control cultures have been added to Figure 5.

Comment 13:

Paragraph starting line 266:  not all of the data is shown in Figure 5 so it is difficult to interpret some of this data (6 hour single species cultures, ghost cells shown in monocultures at different time points, etc).  If the authors have images from all time points (which presumably they do since they discuss what the cells looked like) then it would be helpful to add these to the figure to show the complete morphological patterns.

Authors’ response: thank you; yes, we have images from all time points. However, it seems to us that by placing them in the work, the figure will occupy several pages of the manuscript. Therefore, if you agree, we include the names of morphologically altered cells on the sample photos. We hope that such a procedure will allow for easier interpretation of the obtained results.

Comment 14:

Discussion:  the type 6 secretion system is a major mechanism for antagonism between Gram-negative bacteria.  Pseudomonas and E. cloacae both encode T6SS genes.  Could the authors please comment on whether they think inhibition mechanisms like T6SS are responsible for the decreased viability in the triple species cultures?

Authors’ response: thank you for your valuable comment; the Discussion section has been extended and modified. The type 6 secretion system (T6SS) which is a major mechanism for antagonism between Gram-negative bacteria has been taken into account by us. New articles have also been added to the manuscript. We hope that the manuscript is more valuable now – lines 393-410 and 453-513.

Round 2

Reviewer 3 Report

The authors did a careful job of making changes to the manuscript, which has resulted in an improved manuscript.  I have 2 additional comments:

1) the description of biological vs technical replicates added by the authors is appreciated but but the definition of replicate on lines 108-110 is misleading.  If an experiment was done 3 times each with 6 measurements, it is misleading to say the experiment had 18 replicates as the reader may interpret this as "18 independent biological replicates."  If the experiment was repeated 3 times (meaning it was done on 3 separate days or with 3 different overnight cultures), then there were 3 biological replicates.  If 6 wells were read each time the experiment was done, then there were 6 technical replicates per biological replicate. It is therefore preferred to say "3 biological replicates were done and each had 6 technical replicates."

2) the addition of T6SS as a possibility is nice in the discussion but there does not need to be a page of detail about T6SS systems in Gram-negative bacteria as this is not a paper on T6SS.  Only 2-3 sentences are appropriate in this case.

Author Response

Dear Reviewer,

thank you for the additional comments. We reorganized the manuscript according to your suggestions.

Reviewer comments:

1) the description of biological vs technical replicates added by the authors is appreciated but the definition of replicate on lines 108-110 is misleading.  If an experiment was done 3 times each with 6 measurements, it is misleading to say the experiment had 18 replicates as the reader may interpret this as "18 independent biological replicates."  If the experiment was repeated 3 times (meaning it was done on 3 separate days or with 3 different overnight cultures), then there were 3 biological replicates.  If 6 wells were read each time the experiment was done, then there were 6 technical replicates per biological replicate. It is therefore preferred to say "3 biological replicates were done and each had 6 technical replicates."

Authors’ response:

The description has been reorganized – lines 108-109: Three biological replicates were done, and each had 6 technical replicates

2) the addition of T6SS as a possibility is nice in the discussion but there does not need to be a page of detail about T6SS systems in Gram-negative bacteria as this is not a paper on T6SS.  Only 2-3 sentences are appropriate in this case.

Authors’ response:

The text describing the T6SS system in Gram-negative bacteria has been shortened – lines 452-462. Thus, the number of references has also been changed.